# The Continued Impact of the COVID-19 Pandemic on Pediatric Obesity: A Commentary on the Return to a Healthy New “Normal”

**DOI:** 10.3390/ijerph19095597

**Published:** 2022-05-05

**Authors:** Eileen Chaves, Sheethal D. Reddy, Adelle Cadieux, Jessica Tomasula, Kimberly Reynolds

**Affiliations:** 1Nationwide Children’s Hospital, The Ohio State University, Columbus, OH 43205, USA; 2Children’s Pediatric Institute, Emory University, Atlanta, GA 30322, USA; sheethal.reddy@choa.org; 3Helen DeVos Children’s Hospital, Michigan State University, Grand Rapids, MI 49503, USA; adelle.cadieux@spetrumhealth.org; 4WakeMed Health & Hospitals, Raleigh, NC 27610, USA; jtomasula@wakemed.org; 5Institute on Development and Disability, Oregon Health & Science University, Portland, OR 97239, USA; guion@ohsu.edu

**Keywords:** pediatric obesity, COVID-19 pandemic, obesity guidelines

## Abstract

Two years into this pandemic, mental health symptoms are more prevalent in children and adolescents, routine wellness visits have decreased, individuals and families are experiencing increased stress, and food and nutrition insecurity are on the rise. Pediatric overweight and obesity are yet another health condition that has been impacted by the pandemic. The current commentary aims to (a) summarize a variety of factors contributing to worsening obesity and healthy lifestyle choices in youth throughout the pandemic and to (b) provide recommendations for healthcare providers on navigating this challenge. Specific health behaviors, such as increased sedentary behavior, decreased physical activity, a change to families’ home-food environments, and an increase in sleep dysregulation have contributed to increased weight gain in children and adolescents. As uncertainty continues with the advent of various COVID-19 variants, it remains important to consider how the pandemic has impacted pediatric overweight and obesity.

## 1. Introduction

The COVID-19 pandemic has had an impact on the health behaviors and overall well-being of youth and their families with evidence of increasing risk for pediatric obesity as well [1,2,3]. Halting the spread of the pandemic required sheltering in pace, working from home, remote learning and cessation of social gatherings and extracurricular activities. For many, these changes highlighted pre-existing social inequities. The intention of this commentary is to hone in on the impact of such changes and their specific risk for pediatric obesity. We aim to highlight areas that have potential for intervention by primary care providers. Broadly speaking these areas include food insecurity, sleep regulation, eating in the absence of hunger, safety concerns of diverse youth, diet and exercise. As these issues are not exclusive of one another, our commentary includes a discussion of their interrelatedness and potential methods of holistic intervention. 

*Social Inequities.* Although the relationships are complex, there is a pattern of disproportionate obesity rates among low socioeconomic families, particularly Black and Brown families [3]. The pandemic has shed light on the vast disparities in resources and options available to financially secure families compared to those with less financial stability [4]. Likewise, as families grappled with significant reductions in household income, food insecurity rates were on the rise. Food insecurity is defined as “having inadequate access to sufficient, safe, and nutritious food to meet dietary needs and food preferences for an active and healthy lifestyle” [5]. Tester and colleagues [6] note that while the literature is mixed on the exact association between food insecurity and obesity, both factors commonly occur and negatively affect the health of developing youth.

As youth and their families experienced a wide range of stressors throughout this pandemic, accessing mental health support through the healthcare system has not been equitable. Depending on one’s state of residence, treatment and management of these conditions may not be covered by health insurance, requiring many families to pay out of pocket. The affordability of these interventions is complicated by the noticeable rise in underemployment, unemployment, and food insecurity resulting from the COVID-19 pandemic [7,8]. The COVID-19 pandemic intensified food insecurity, as children and families initially lost access to free or subsidized healthy school meals, families lost income, and supply chain issues reduced options in stores. These issues were amplified in lower income and predominantly minority areas, placing more youth at risk for unhealthy eating habits [9]. In response, many school districts began offering free lunch and breakfast for all students regardless of income status [3,10]. Lack of health insurance coverage results in less access to quality care, such as multidisciplinary pediatric weight management clinics, and delays in the care of chronic health conditions such as obesity. The COVID-19 pandemic has also resulted in a significant reduction in access to collaborative medical groups in treating pediatric obesity. For instance, Johnson and colleagues [11] reported nearly 91% of pediatric sleep centers stopped or significantly reduced in-laboratory studies. Likewise, community-clinic partnerships were halted due to COVID-19 restrictions (e.g., community-based group fitness, increased demands on local food banks). 

*Stress and Eating in the Absence of Hunger.* The impact of stress on metabolic health and weight status is robust [12,13], and the pandemic clearly exacerbated familial and individual stress for many. Coincidentally, while much of the country remained in some form of lockdown or quarantine in early 2020, the murders of three Black people (Ahmaud Arbery, Brionna Taylor, and George Floyd) in February, March, and May of 2020, respectively, drew international attention and outrage. Undoubtedly, these events created additional stress within some subgroups of the American population, particularly Black and Brown individuals. Ang [14] demonstrated that police killings, even those that went unreported in the media, resulted in increased absenteeism and emotional disturbances as well as decreased grade point averages for high school students who lived near where the incidents occurred. Additionally, these negative effects disproportionally affected Black and Hispanic students for several years after these events, which was evident by lower high school graduation and college enrollment rates among students who were ninth graders when the police killings happened [14].

Eating in the absence (EAH) of hunger is a common short-term but poor long-term coping strategy. There is evidence that it may be more prevalent in low socioeconomic households [10]. EAH encompasses eating that is mindless and impulsive (“because it’s there”) as well as eating that is intended to soothe or alleviate painful feelings (“stress eating”). Unlike binge eating, EAH can describe eating with or without loss of control. Anecdotally, there is evidence that EAH episodes became more frequent as people were spending more time at home. Increased eating episodes likely correlate with increases in BMIs during this time, although additional data would be needed to confirm this association.

A study in Uruguay [15] of changes in eating patterns during the pandemic indicated some youth increased food intake out of boredom and/or anxiety and other youth decreased food intake due to sadness. However, the type of food varied, with some participants indicating increased unhealthy food options and others indicating healthier home-cooked meals [16]. In Canada, researchers found that during the pandemic, youth were eating more snack foods even though they were eating out less and caregivers were making more meals at home [17]. Although these studies were not completed in the US, similar stories have been reported around the country with some families working on making healthy home-cooked meals, while other families have struggled more with food insecurity and having access to healthier food options. 

Finally, change in work and school resulted in decreased caregiver supervision within some families. Although parental monitoring has not been consistently shown to impact eating behavior in youth [18], the significant impact the pandemic has had on many caregivers’ ability to monitor their youths’ eating, especially eating in the absence of hunger, may play a role in youths’ eating habits.

*Risks for Diverse and Special Needs Youth.* There is growing evidence that a significant portion of adolescents identifying as transgender or gender non-conforming are overweight [19]. For vulnerable youth within the LGBTQ+ community, stay-at-home orders may have restricted access to in-person social support and increased risks for domestic violence and abuse. From a practical perspective, remote learning removed the safety net of mandated reports such as teachers and therapists [20,21].

Prior to the pandemic, families raising children with intellectual impairments, those on the autism spectrum, with trisomy 21 (Down syndrome) or Prader–Willi syndrome (PWS) may have relied on a network of family, friends, therapists, teachers, and support groups for care, instruction, and day-to-day assistance. During the pandemic, caregivers of youth with special needs have had to assume not only the roles of teacher, babysitter, and parents but potentially the roles of occupational, speech, and physical therapists. An added responsibility for caregivers for children with active hyperphagia is managing their child’s diet [21,22,23,24]. Children with hyperphagia may attempt to obtain food outside of scheduled meals and snacks, eat large portions rapidly and eat in secret [23]. Preoccupation with food and hyperphagia, common features of PWS and some children with trisomy 21, may be harder to manage with parents having the added demand of dividing their time between work, supervising academic demands, and reduced childcare help while working from home [25,26]. 

## 2. Healthy Habits: Sedentary Behavior, Physical Activity, Sleep and Nutrition

Compounding the challenges of the pandemic, youth in weight management programs likely were unable to receive intervention as usual during the height of the COVID-19 pandemic. Treatments interventions were halted, modified (e.g., virtual), and at times put on hold. For families of youth with these conditions, weight maintenance is difficult even in the best-case scenarios; hence, we would anticipate even greater weight gain during this particularly stressful time. 

Preliminary studies have indicated an increase in sedentary behaviors during the pandemic as children began using electronic devices for remote learning and leisure activities. Sedentary behaviors increase the risk for obesity and cardiometabolic disease in adults and youth [27]. A recent study in China found that during the COVID lockdown in their country, youth had decreased physical activity and increased screen time. In addition to the changes in youth activities, the study found that BMI increased as did the rate of overweight and obesity in youth [28]. A Canadian study also showed increased screen time and decreased physical activity during the beginning stages of the pandemic [3].

American Academy of Child and Adolescent Psychiatry (AACAP) guidelines recommend ≤2 h of recreational screen time per day for 5–17-year-olds. However, even before the COVID-19 pandemic nearly half of North American children (age 8–12 years old) spent 4–6 h per day engaging in screen time (e.g., watching TV, playing video games, scrolling through social media) [27]. Many children have reported that electronic devices are one of the few ways to safely connect with family and friends during the pandemic; however, increased online activity places youth at risk for behavioral health symptoms [29] and reductions in physical activity.

The disruption in children and adolescents’ schedules by the pandemic has shown a dramatic decrease in levels of physical activity [28]. Prior to the COVID-19 pandemic, less than 10% of school-aged (5–17 years) youth achieved recommended daily amounts of physical activity: specifically, 60 min of moderate-to-vigorous physical activity per day [28]. Additionally, 19% of children with disabilities met the recommendation of 60 min of physical activity daily [30]. Many parents also began to experience increased demands on their time, as many juggled working from home with also helping to teach their children or continuing to work at essential jobs while their children stayed home to engage in virtual schooling alone. This often resulted in challenges in finding ways to keep their children physically active [31]. As gyms closed and youth sports halted, children had fewer opportunities for structured physical activities. With the closure of school buildings and the discontinuation of extracurricular activities, youth no longer had access to the social interactions that these activities provide. 

There is increasing research linking a lack of sleep with an increase in pediatric obesity. Studies have shown that sleep restriction often leads to weight gain in humans [32,33,34,35]. During COVID-19, altered school and work schedules resulted in children having less sleep overall [17] and possibly later-than-usual bedtimes as well as sleeping later during the day [27]. Youth also reported increased anxiety about the pandemic interfering with their ability to fall asleep in a large community sample of Chinese adolescents [36]. Increased food intake in response to sleep dysregulation is thought to be the current mechanism of action that leads to weight gain. In studies where participants have engaged in voluntary sleep restriction, they were more likely to engage in increased snacking, eating more frequent meals, and a preference for energy-dense, nutrient-poor foods [35], which are all behaviors that place individuals at increased risk for overweight and obesity. 

The home-food environment changed for many families during the pandemic. Lockdown measures reduced the frequency that many families engaged in grocery shopping, reducing the purchase of fresh fruits and vegetables [37] and possibly increasing the purchase of processed foods that have a longer shelf-life. Pandemic-related financial strain coupled with concern for shopping in large supermarkets may have led families to rely on nearby corner stores or drive-through fast food chains more often than usual. These changes in eating habits may have affected lower-income families more as they are more likely to experience “food deserts” (i.e., limited access to fresh foods and whole grains) and “food swamps” (i.e., areas highly populated with fast food and convenience stores), which have been strong predictors in adult obesity rates [38].

*Interrelatedness of healthy habits and the larger environment.* As mentioned in the Introduction, these various aspects of human habits are not mutually exclusive. When remote and asynchronous learning and work became more the norm during the pandemic, sleep schedules shifted, resulting in less time spent in restful sleep at night in adults [39] and possibly youth, although data are not available to clarify [27]. Staying awake late into the night has been associated with increased snacking [35], and a shift in sleep thus could explain some changes in eating habits. Combined with a lack of structured activity, these changes may easily lead to weight gain. Additional familial stressors including financial hardships, poor coping skills, and food insecurity may lead to further eating dysregulation. As such, an ideal intervention would be one that targets multiple domains at once. 

## 3. Moving Forward

The challenges of the pandemic on healthy lifestyle choices within the last two years are clear and numerous. An appropriate next question is, “How do we move forward from this point?” First, we must recognize that despite encouraging increases in vaccination rates and return to school efforts, our new normal will be different in some important ways. We will continue to experience iterations of safety precautions and fluctuating access to sports teams, gyms, social gatherings, recess, dinner parties, etc. Our medical and therapeutic approach will need to remain nimble and tailored. 

A first suggestion for moving forward is to consider families’ functioning within the context of Maslow’s hierarchy of needs. The balance of encouraging healthful behaviors while prioritizing safety and stability requires careful attention to one’s communication approach to avoid a victim-blaming attitude that is already often imposed on marginalized communities. The pandemic has exacerbated psychosocial stressors such as food and housing security, especially for the communities who were already the most marginalized. It is critical that families are supported in identifying priorities for whole family health while not overwhelming them with new goals or tasks. Thus, there is an increased need for grace and flexibility in the provider role. For example, we may be more pressed to focus on basic needs such as food security or maintaining a daily schedule while children are engaged in hybrid schooling before engaging in coaching on healthy snacking habits. 

Second, telehealth has become fairly ubiquitous during the pandemic; most specialty and primary care providers have now had the occasion to practice this way. However, insurance coverage of these services is at risk as the pandemic wanes and more traditional practices and policies return. Although the rapid introduction of telemedicine occurred as a crisis response, telehealth visits have provided an important access point to families who need medical care, especially those who may have difficulty with transportation, distance to care, or taking time from work to attend appointments in person. A great deal of pediatric obesity treatment is centered around behavioral change. In many cases, virtual care is well suited for conducting appropriate assessment and delivering guidance to families in this domain [36]. In fact, some providers have experienced improvement in their care when patients are home, because they can easily reach into their pantry to show a dietitian what is there, ask about a particular food label, or share a medication bottle. Because physical examinations, physical therapy evaluations, and some level of behavioral observation is necessary in person, a hybrid model of in-person and virtual care may ultimately be the best balance for many families. We would recommend providers consider incorporating telemedicine into their obesity treatment approach and that they engage in advocacy to ensure that cost-reimbursement for this important service continues into the future.

With regard to using food as a coping mechanism, depending on the severity and frequency of the behavior and its impact on a child’s weight status, a referral for cognitive–behavioral intervention may be warranted. Coping skills training and stress management interventions have been shown to be effective when delivered via a virtual format [40]. 

Next, families and providers are encouraged to take advantage of this season in particular—the season of change. As our schedules and routines are shifting, yet again, from spring to summer and from virtual school to hybrid or in-person school, it is an apt moment to implement structures and routines that are healthful. One key to maintaining any behavioral health change is to make it routine and in sync with other activities that we are confident will occur. Specific strategies that can be applied to support daily routines and structure are outlined below. Please also refer to the relevant article published by the American Academic of Pediatrics: “Helping kids stay active & eat healthy during COVID-19” published online in December 2021 [41]. 

Daily meal schedule: Posting a visible schedule of when meals and snacks will occur can help reduce “grazing” or unnecessary snacking throughout the day. The expectation that all food is eaten at the table (not in front of the computer or other screen) can also help with this. Visual cues such as a sign indicating when the kitchen is open or closed can also be useful.Boredom list: Many children (and adults) wander into the kitchen when they are bored or distressed. Creating a visual list of alternatives for each of these scenarios can serve as a reminder of other things to do. Have children and parents work together to create a list of ideas they can engage in when bored and post it near the kitchen, at eye level. Including a mix of silly ideas (play a safe trick on mom, find as many triangles as you can and take photos) and productive ones (build a Lego castle, read a book, clean the bathroom) can make this more appealing for some children. Remove the temptation: Since families are spending more time at home, removing the tempting and non-nutritious foods is often the simplest and most effective answer. If there are no sodas in the house, there will be no sodas to drink. Encourage families to decrease access to these foods by purchasing smaller portions/packages of high calorie, nutrient-poor foods or limiting them altogether. Build activity into the day: Attending school and public outings/gatherings included built-in reasons to move from one space to the next. These natural opportunities to move the body can be built into a home-based day as well. Families can be encouraged to identify chores, errands, or other reasons to pair an existing requirement with movement or outdoor time. Building physical activity and movement into the day, as an expectation as natural as getting dressed or attending school, will make it more likely to happen. Modeling, modeling, modeling: Research is clear that children are most likely to engage in long-term behaviors that are modeled by their caregivers (regardless of what was verbally taught). The important role of parent health behaviors cannot be overstated. Parents should be encouraged to join in and model any health behavior goals they have for their children. In this way, they can provide important company and motivation to engage in healthy behaviors, support problem solving by directly experiencing the challenges of health behaviors, and as noted, provide an example that is observable to the child. 

Providers are also encouraged to stay informed about and support policies that enhance family access to basic needs such as food and housing access. Medical specialists are not trained about the importance of their roles in advocacy and policy work and, therefore, are often not engaged in these important processes. However, the clear intersections between medical health and psychosocial resources call for an increasingly engaged medical work force.

## 4. Conclusions

Families and providers are encouraged to “celebrate wins” and focus attention on the healthful behaviors rather than perseverating on what may not be going well. Emphasizing the maintenance of existing healthy habits may be particularly helpful for families experiencing significant financial or other pandemic-related stressors. As we emerge into a “new normal”, encouraging families to focus on concrete modifications may have more success. This might include focusing on the achievement of a specific goal such as trying a new low-calorie, nutrient-dense food that day, finding an affordable and healthy food option, drinking water rather than an alternative, or stepping away from the computer for five minutes to move. Increases in healthy behaviors are more likely when we are praised and rewarded for small successful efforts rather than chastised for failures. All of us may do well to increase our gratitude and praise toward ourselves and one another.

## Data Availability

Not applicable.

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
