# Peer review of "The Continued Impact of the COVID-19 Pandemic on Pediatric Obesity: A Commentary on the Return to a Healthy New “Normal”"

_ijerph, 2022, doi:10.3390/ijerph19095597_

Round 1

Reviewer 1 Report

In this commentary, the authors aimed to compile the factors contributing to increased risks of childhood obesity during the Covid-19 pandemic. In the light of this synthesis, they offered several recommendations for moving forward. Although this is an interesting topic, I suggest making major corrections throughout the manuscript as explained below.

Major points

  • I think the introduction is too long and unfortunately does not directly relate to what has been covered in this paper. Please summarize the general impact of the pandemic on youth. Then introduce obesity and the problem of weight maintenance in youth so that you can link the two sections and introduce the reader to the content of this commentary. "Why do you think the Covid-19 pandemic would have a specific impact on childhood obesity and weight maintenance in children?"
  • How health behaviors interact with each other is, unfortunately, missing in this paper. For instance, how increased sedentary behaviors (SB) and decreased physical activity would affect sleep? Then how sleep would affect circadian rhythm and metabolism or anxiety/stress.

This could give an overview of the synthesis. I suggest adding a commented scheme to present this overview (please include SB, physical activity, sleep, circadian misalignment, dietary intake as well as all health behaviors mentioned in this paper)   

Please see below for specific comments.

Abstract:

Line 12-17: The abstract could do with a more rounded approach to better reflect the content of this commentary. For instance, the most important health behaviors affected by the pandemic and major recommendations should be added.

Line 18-20: During which period? please specify.

Line 118: please remove the reference and use accurate citation. Examples are given below.

  • Martinez-Gomez, D., Eisenmann, J. C., Healy, G. N., Gomez-Martinez, S., Diaz, L. E., Dunstan, D. W., ... & AFINOS Study Group. (2012). Sedentary behaviors and emerging cardiometabolic biomarkers in adolescents. The Journal of pediatrics, 160(1), 104-110.
  • Rendo-Urteaga, T., de Moraes, A. C. F., Collese, T. S., Manios, Y., Hagströmer, M., Sjöström, M., ... & HELENA Study Group. (2015). The combined effect of physical activity and sedentary behaviors on a clustered cardio-metabolic risk score: The Helena study. International journal of cardiology, 186, 186-195.

Line 169: References specific to young populations should be provided and detailed

  • Duraccio, K. M., Krietsch, K. N., Chardon, M. L., Van Dyk, T. R., & Beebe, D. W. (2019). Poor sleep and adolescent obesity risk: a narrative review of potential mechanisms. Adolescent health, medicine and therapeutics, 10, 117.
  • Duraccio, K. M., Whitacre, C., Krietsch, K. N., Zhang, N., Summer, S., Price, M., ... & Beebe, D. W. (2021). Losing sleep by staying up late leads adolescents to consume more carbohydrates and a higher glycemic load. Sleep.
  • Beebe, D. W., Simon, S., Summer, S., Hemmer, S., Strotman, D., & Dolan, L. M. (2013). Dietary intake following experimentally restricted sleep in adolescents. Sleep, 36(6), 827-834.

Reviewer 2 Report

Thank you for the material sent. This work is interesting and clearly summarizes the threats of the COVID-19 pandemic in the context of socio-economic risk factors for the development of obesity.

The advantage of the work is simple and practical advice that can be implemented by medical specialists. 

Reviewer 3 Report

The presented manuscript is a popular-scientific review of unsystematic literature of little scientific value.

The weakness of the study includes:

- lack of description of the methodology of reviewing sources of thematic knowledge and qualifications in accordance with the applicable PRISMA canons

- the description of individual factors of the article is sketchy and not exhaustive

- after reading the article, the Reader does not learn anything new

- although the discussed topic concerns the COVID-19 pandemia, a significant part of the sources used come from the period before the pandemic

- a small number of found and used items of the bibliography

In view of the arguments presented above, I do not recommend the article for further stages of the journal publishing process.

Round 2

Reviewer 1 Report

Major points:

First, I still think the introduction is too long and does not directly relate to what was covered in this paper. 
Please keep it short, clear, and focused on the question treated in this commentary.
Could you outline the objectives of the commentary at the end of this section?

Second, the authors state that they have addressed how health behaviors interact with each other.  Could you highlight these parts? After reading the article, I can't find a single example.

It is regrettable that the "overview scheme" suggestion was not considered.

Minor points:
Line 144-145: “Sedentary behaviors increase the risk for cardiometabolic disease in adults. Similar findings are emerging in youth [11].”
The reference is not accurate. The paper by bates et al. (2020) deals with the effect of the pandemic on physical activity, sedentary behavior, and Sleep in youth. References treating the effect of sedentary behaviors on the risk of cardiometabolic disease in youth were given in my previous report.

Reviewer 3 Report

The corrections introduced by the authors do not clarify the doubts when reading the manuscript.
In the title of the article, please enter the type of publication (commentary).
The introduction also requires clarifying the scope and type of the commentary. It is also worth mentioning at this point the scope of bibliographic sources from which these commentaries were made. In some paragraphs, eg.  Increase in sleep dysregulation - their commentary is based on two source reports.
